# Experimental evidence of quantum radiation reaction in aligned crystals

Tobias N. Wistisen [1], Antonino Di Piazza[2], Helge V. Knudsen[1] & Ulrik I. Uggerhøj[1]

Quantum radiation reaction is the influence of multiple photon emissions from a charged particle on the particle's dynamics, characterized by a significant energy-momentum loss per emission. Here we report experimental radiation emission spectra from ultrarelativistic positrons in silicon in a regime where quantum radiation reaction effects dominate the positron's dynamics. Our analysis shows that while the widely used quantum approach is overall the best model, it does not completely describe all the data in this regime. Thus, these experimental findings may prompt seeking more generally valid methods to describe quantum radiation reaction. This experiment is a fundamental test of quantum electrodynamics in a regime where the dynamics of charged particles is strongly influenced not only by the external electromagnetic fields but also by the radiation field generated by the charges themselves and where each photon emission may significantly reduce the energy of the charge.

---

[1] Department of Physics and Astronomy, Aarhus University, Ny Munkegade 120, 8000 Aarhus, Denmark. [2] Max Planck Institute for Nuclear Physics, Saupfercheckweg 1, 69117 Heidelberg, Germany. Correspondence and requests for materials should be addressed to T.N.W. (email: tobiasnw@phys.au.dk)

A complete understanding of the dynamics of charged particles in external electromagnetic fields is both of purely theoretical interest and of practical importance. In fact, it has fundamental consequences in several branches of physics, spanning, for example, from pure particle physics and accelerator physics to plasma physics and astrophysics. Since accelerated charges, electrons for definiteness, emit electromagnetic radiation, in the realm of classical electrodynamics a self-consistent equation of motion of the electron in an external electromagnetic field must take into account the resulting loss of energy and momentum[1,2]. However, although the inclusion of energy-momentum loss in the determination of the electron's trajectory has important practical implications, it is also intimately related to the nature of the electron and of its electromagnetic field, and it thus represents by itself an outstanding problem in fundamental theoretical physics. In fact, the self-consistent approach to tackle this problem is based on the system of equations describing the coupled dynamics of the electron (Lorentz equation) and of its own electromagnetic field (Maxwell's equations) in the presence of the given external field[3]. The electron is driven by both the external electromagnetic field and its own electromagnetic field (radiation reaction), and a proper expression of the latter in terms of the dynamical quantities characterizing the electron allows for constructing an equation containing only these quantities (the Lorentz-Abraham-Dirac (LAD) equation). However, the LAD equation turns out to be plagued by serious inconsistencies like the admittance of runaway solutions, with the electron acceleration diverging exponentially in time even if the external field identically vanishes[3–7]. Thus, the problem of radiation reaction is extremely important not only from a practical point of view but also from a fundamental perspective. Within the realm of classical electrodynamics, Landau and Lifshitz[2] have shown that the LAD equation can be consistently approximated by another equation, known as Landau-Lifshitz (LL) equation, which is not plagued by the above-mentioned inconsistencies of the LAD equation (see Eq. (1) in Supplementary Note 1). In ref. [8] it has then been shown that all physical solutions of the LAD equations are also admitted by the LL equation. However, a full understanding of the above-mentioned inconsistencies is, in fact, possible only within the more fundamental quantum theory, quantum electrodynamics (QED). Analogously, as in classical electrodynamics, the complete inclusion of radiation reaction effects would amount in solving exactly the quantum coupled equations of motion and to account for all processes initiated with an electron in the initial state in the presence of the background field and for the corresponding radiative corrections. Within QED the notion of background electromagnetic field refers to the part of the total electromagnetic field, which is so intense that it does not need to be quantized because it contains a sufficiently large number of coherent photons and is approximately unaltered during the quantum process under investigation[9]. The background field is then treated as a classical field, whose time evolution is given. In the present case the background field is the total field produced by all the atoms of the crystal. In its own generality the problem of quantum radiation reaction is thus multiparticle because, unlike in classical electrodynamics, the radiation (photons) emitted by the electron interacts with the background field and transforms into electron-positron pairs. One of the main reasons why such an old, fundamental, and outstanding problem as the radiation reaction problem is still unsolved, relies on the difficulties in detecting it experimentally. The rapid development of laser technology has renewed the interest in this problem because the strong fields provided by intense laser facilities may allow for the experimental measurement of radiation reaction effects (we refer to the review[10] for papers until 2012 and we also mention the recent studies[11–21]). In that context, the relation between radiation reaction in QED and the emission of multiple photons in a regime where quantum recoil is substantial has been pointed out in ref. [22]. In ref. [23] we have realized that the strong electric fields in aligned crystals may be also suitable for measuring radiation reaction effects and test the LL equation. In an aligned crystal, in fact, under suitable conditions identifying the so-called channeling regime, an electric charge also oscillates similarly as in a laser field and may thus radiate a substantial fraction of its energy. In the following, we report results of an experiment aiming at measuring radiation reaction effects on the photon emission spectra of ultrarelativistic positrons crossing an aligned crystal. Each positron is found to emit several high-energy photons with non-negligible recoil such that both radiation reaction and quantum effects dominate the positron dynamics under the conditions of the experiment. The measured spectra are, in fact, best explained by a quantum model of radiation reaction.

## Results

**The experiment and the regime of radiation emission.** In the experiment described below, ultrarelativistic positrons cross a Si crystal in the channeling regime. The experiment has been performed at the Super-proton-synchrotron North Area facility at CERN employing positrons with incoming energy of $\varepsilon_0 = 178.2$ GeV and two Si crystals with thickness 3.8 and 10 mm, respectively, aligned along the <111> axis. Since the background crystal field is purely electric, the quantum parameter is defined as $\chi = \gamma E/E_{cr}$, where $E$ is a measure of the crystal field amplitude, $\gamma$ is the initial Lorentz $\gamma$-factor of the positrons, and $E_{cr} = m^2 c^3/\hbar|e| = 1.3 \times 10^{18}$ V/m is the critical field of QED[9,24–26]. As we will see, under the conditions of the experiment, the parameter $\chi$ is of the order of unity or less. Consequently, both radiative corrections and pair production give negligible contributions[26,27] and the inclusion of quantum radiation reaction corresponds to taking into account the multiple emission of photons by each positron. From a theoretical point of view, the problem would still be unsolvable because of the technical difficulties in calculating multiple photon emission probabilities including exactly the effects of the background field and of the complex space dependence of the crystal field itself. Another feature of our experiment, however, is that the parameter $\xi = p_{\perp,max}/mc$, where $p_{\perp} = \gamma m v_{\perp}$ is the transverse positron momentum, is larger than unity for most of the positrons[26]. In the ultrarelativistic regime the parameter $\xi$ also represents the maximum angular deflection during the electron's motion from its average direction, divided by the characteristic angle of radiation $1/\gamma$. Although, in the regime of the experiment the condition $\xi \gg 1$ (synchrotron limit) is not always fulfilled, we have been able to reproduce most of the experimental results by employing a feasible approach based on the synchrotron limit. In this limit, in fact, multiple photon emissions can be approximated as the consecutive emissions of several photons and the single-photon emission probability is formed on a length much smaller than the typical length where the background field varies[26]. Thus, for ultrarelativistic positrons as in our experiment, the expression of the emission probability for a constant crossed field (CCF) could be used (see Eq. (4) in Supplementary Note 1), because ultrarelativistic particles approximately see an arbitrary field as a plane wave in their instantaneous rest frame. In conclusion, under the conditions of our experiment, from a quantum point of view, the positrons being ultrarelativistic propagate as pointlike classical particles according to the Lorentz equation[26] and, in a pure random way, emit photons with probabilities and energy distributions calculated according to the quantum formulas. In this respect, our model is semiclassical in nature. In the classical limit, when the

quantum parameter $\chi$ is much smaller than unity, the probabilities reduce to the corresponding classical quantities (that is, the probability times the photon energy coincides with the classical intensity of radiation) and the recoil due to each photon emitted is on average much smaller than the positron energy such that the stochastic emission of photons essentially becomes a continuous emission of radiation.

In our experiment, the dynamics of the positrons is characterized by $\chi \lesssim 1.4$ and $0.7 \lesssim \xi \lesssim 7$ such that one is in the quantum regime and the field is either classically strong or in an intermediate regime below the CCF regime $\xi \gg 1$. The measured photon emission spectra show features that can only be explained theoretically by including both quantum effects related to the stochasticity of photon emission and related to the fact that the quantum probabilities and photon energy spectra are different than in the classical limit $\chi \ll 1$ and, additionally, radiation reaction effects stemming quantum mechanically from the emission of multiple photons and from the inclusion of the energy-momentum loss at each emission for the determination of the subsequent positron dynamics. Several experiments have studied the emission of radiation in crystals in the quantum regime, mostly in thin crystals to avoid pile-up effects in the calorimeter, that is, the emission of multiple photons by a single particle has been avoided, such that only quantum effects have been measured but not radiation reaction effects. Due to pile-up, in fact, only the sum of the energies of all the photons emitted by each charged particle is measured in such calorimeter experiments, which prevents the possibility of reconstructing the single-photon spectrum (in ref. [28], such a pile-up effect can be seen). Here in the present experiment we have instead employed a thin converter foil and a magnetic spectrometer to obtain the single-photon spectrum (Fig. 1). Therefore, in the radiation reaction regime where many photons are emitted by a single positron, the current experiment clearly provides more information on the dynamics of the positrons and a stronger test of the theory than previous experimental campaigns. The results of the current experiment should also be contrasted with the effects seen in radiation emission in synchrotron facilities. In synchrotrons, the magnetic field and the electron energies are such that the quantum nonlinearity parameter $\chi$ is always much smaller than unity and the emission of radiation is appropriately described by the classical synchrotron spectrum formula found in ref. [1]. At each turn the energy-momentum loss is reduced by a predictable amount in both longitudinal and transverse directions. The electrons are then re-accelerated in the longitudinal direction using RF cavities resulting in the particles now having smaller transverse angles. When this damping effect has taken place over a long period of time and the oscillation amplitudes in the

synchrotron have become significantly dampened, a quantum excitation effect becomes important[29–32], which puts a limit on the lowest emittance, which can be achieved. In the case of our experiment the quantum effects are different. In contrast to the synchrotron, first we cannot use the classical formula of synchrotron radiation because this would give completely inaccurate results as $\chi$ even exceeds unity in our experiment[33]. Instead we must use the formula found in refs. [9,26,27] derived from the solution of the Dirac equation. Second, since the energy radiated at each emission, in the case of our experiment, is significant as compared to the particle energy, the momentum of the particle after emission is no longer predictable, that is, one positron does not lose the same amount of energy as the next due to the inherent stochasticity of quantum mechanics. Third, in our experiment radiation reaction is in the full quantum regime especially at the beginning, when the positron energy is maximal, before the multiple recoils reduce it. Finally, the fact that each positron undergoes multiple recoil with energy losses comparable with its own energy implies that on a timescale of picoseconds, its energy is so significantly altered by the radiation itself that its energy emission spectrum is strongly affected. Such an effect could not be in principle observed in a synchrotron where the energy loss occurs over much longer timescales and the very functioning of the machine over multiple turns of the electrons requires a correction of the trajectory of the electrons, which in turn prevents observing radiation reaction effects on the emission spectrum. It is worth observing in this respect that even in the optimistic case of building a synchrotron with LHC strength dipole magnets of around 8 T, electrons of an energy of 281 TeV would be required to reach a unit value of the quantum parameter, which is beyond any foreseeable high-energy accelerator.

In Fig. 1 a schematic of the experimental setup is shown. In Fig. 2a we show the experimentally obtained counting spectra for the background case, when no crystal is present, for the random case when the crystal is present but not aligned with respect to the positron beam, and for the align case, when the crystal's <111> axis is aligned with the positron beam. In Fig. 2b we show a comparison of the experimental and the theoretical results in the amorphous case. The theoretical, simulated curves are denoted by sim (see also Supplementary Note 2). In the vertical label of this plot $X_0 = 9.37$ cm is the radiation length of Si. In the random orientation the radiation emission is the well-understood Bethe-Heitler bremsstrahlung[34] and the agreement here therefore shows that the simulation of the setup is accurate. The result in the random orientation was used as a way to normalize the theoretical results to the experiment by a scaling factor. This is necessary since the efficiency of the setup depends not only on the geometry of the setup, multiple scattering, etc. but also on the inherent efficiency of the MIMOSA detectors.

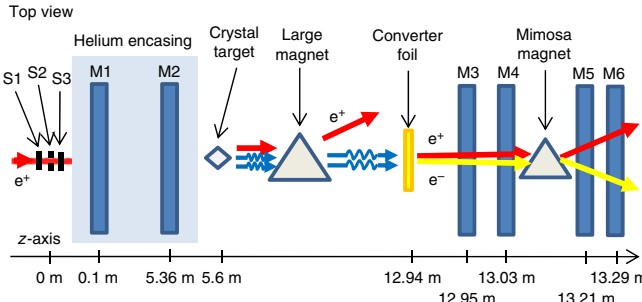

**Fig. 1** Experimental setup. A schematic representation of the experimental setup in the H4 beam line in the SPS NA at CERN. The symbol $S_j$, with $j = 1$, 2, 3 denotes the scintillators and the symbol $M_i$, with $i = 1, ..., 6$, denotes MIMOSA position sensitive detectors

**Comparison between experiment and theoretical models**. We have considered four different theoretical models to compare with the experimental results. In general, we point out that since in our experiment multiple photon emission is an essential feature, the dependence of the photon spectra on the density and the thickness of the target is nonlinear, such that an approach based on the introduction of a cross section like in ref. [35] would be inadequate here. These models are described in the section Materials and Methods in Supplementary Note 1, and, depending on which effects they include, are indicated as classical plus radiation reaction model (CRRM), semiclassical plus radiation reaction model (SCRRM), quantum plus radiation reaction model (QRRM), and quantum with no radiation reaction model (QnoRRM). In Fig. 3 we show the result of such a comparison in

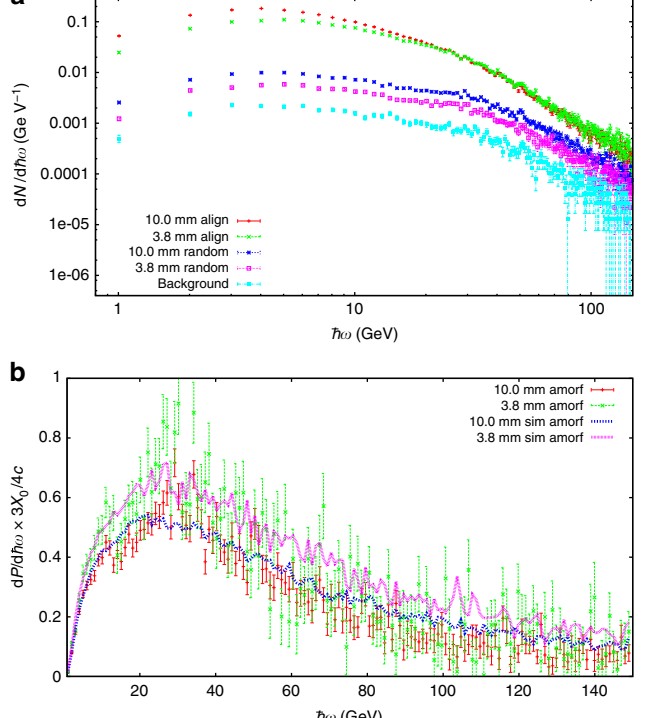

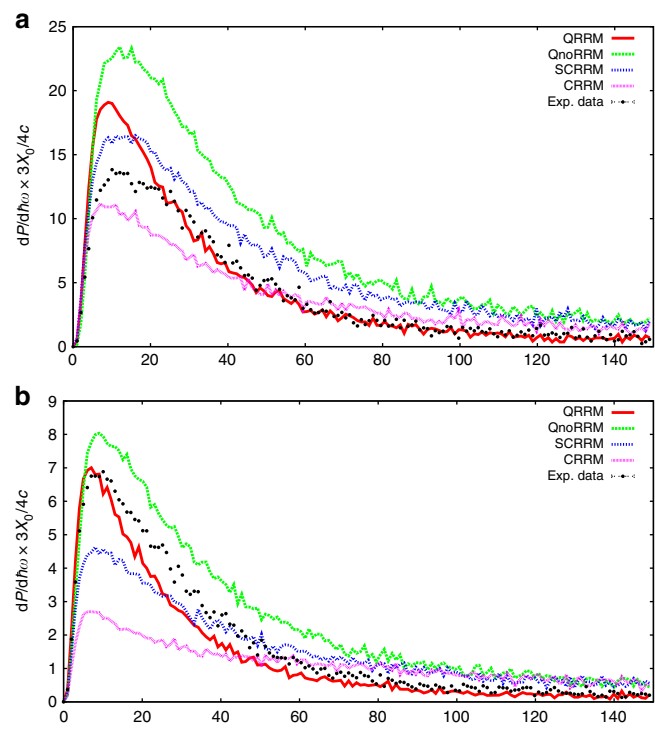

**Fig. 2** Experimental and simulated counting spectra in random orientation. Photon-counting spectra per single incoming positron for the two crystal thicknesses indicated in the text in the aligned and the random case along with a measurement of the background radiation (**a**). The background subtracted power spectrum in the random orientation is compared to simulations (**b**). The error bars are due only to the statistical counting error in each bin

**Fig. 3** Experimental and simulated power spectra. Background subtracted power spectra in the aligned case for two crystal thicknesses: 3.8 mm (**a**) and 10.0 mm (**b**). The experimental data are compared to the four different theoretical models, described in Supplementary Note 1, after being translated through the simulation of the experimental setup. The error bars are due only to the statistical counting error in each bin

the cases of the 3.8 mm crystal (a) and the 10.0 mm crystal (b). As we have anticipated, among the four models described, only the QRRM can be considered in reasonable agreement with the experimental data, indicating the importance of including both quantum and radiation reaction effects in the modeling. In Supplementary Note 3, we show the calculated values of the reduced chi-squared statistic to show the goodness of fit of the various models. We should keep in mind that the chi-squared statistic tests whether the discrepancy between a theoretical curve and an experimental one is of statistical nature or not. In our case, we already know a priori that all of the models for different reasons may not agree with the experimental results (in the QRRM the reason is related to the validity of the CCF approximation). The conclusion indeed is that the discrepancy is not explained by statistical fluctuations, and that in the 3.8 mm case the CRRM is the best model, followed by the QRRM, whereas in the 10.0 mm case the CRRM is the worst model while the QRRM is the best one. In this respect, it also makes sense to consider the overall set of data combining the 3.8 mm case and the 10.0 mm case, and the result is that the QRRM is overall the best one to describe our experimental data. The fact that the classical model works better in the 3.8 mm case does not have to surprise because although the classical model obviously does not include quantum effects, it does not require the CCF approximation (in this respect it is exact). Therefore, we have two competing effects, which render preferable either the quantum model (based on the CCF approximation) or the classical model (independent of the CCF approximation). Moreover, in the 10.0 mm case and in the CRRM, the positrons quickly radiate at low photon energies, due to the larger initial power emission. This causes saturation effects

in the experimental detection and therefore makes the spectrum according to the CRRM to significantly differ from the experimental one. As we have also hinted, the remaining discrepancy can likely be attributed to the use of the CCF approximation in regions at the limits of its applicability. In Fig. 1 of the Supplementary Materials we have shown the difference between the CCF approximation and a more accurate approach using the general formulas of Baier et al.[26] whose numerical procedure is outlined in ref. [36] in the thin crystal case. This shows that the CCF approximation overestimates the emission at low photon energies in this case. It is not clear how this approach can be extended to a thick crystal and to the best of our knowledge, no complete theory of quantum radiation reaction, valid in all regimes, has yet been devised as it would essentially imply an exact computation of the emission probability of an arbitrary number of hard photons. While the CCF approximation is the best theory we can apply to a thick crystal at the moment, the problem seen in the case shown of a thin crystal, see Supplementary Figure 1, must persist in the case of thicker crystals as used in our experiment.

For the sake of completeness, in Fig. 4 we show the positron power spectra according to the four mentioned theoretical models before the translation based on the simulation of the setup has been carried out. Here it is seen that for both thicknesses the curves corresponding to the QnoRRM are the same but that this is not the case after the translation is carried out (Fig. 3). The main reason for this is that the efficiency of the experimental setup depends on the total number of produced photons. This effect becomes severe when the number of photons that can convert in the foil becomes appreciable compared to ~26, considering the 5% of the radiation length $X_0$ converter foil such

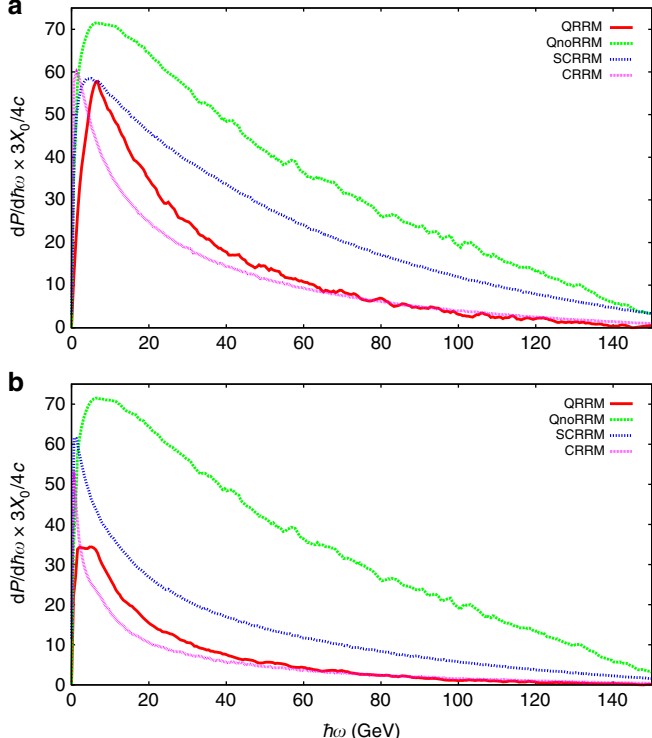

**Fig. 4** Theoretical power spectra. Power spectra calculated according to the four different theoretical models described in Supplementary Note 1, for the two crystal thicknesses: 3.8 mm (**a**) and 10.0 mm (**b**)

that multi-photon conversion becomes likely. In such events the original photon energy cannot be found and is thus rejected (this also shows the necessity of doing such a simulation of the experimental setup). It is seen in the 3.8 mm case that there is a qualitative agreement between Figs. 3 and 4 in the relative sizes of the spectra compared to each other. However, in the case of the 10.0 mm crystal it is seen, for example, that the spectrum corresponding to the QRRM model is higher than that corresponding to the SCRRM in Fig. 3, whereas the opposite occurs in Fig. 4. This is possible due to the many more soft photons being predicted in the SCRRM calculation than in the QRRM, which lowers the translated spectrum because of the discussed rejection of multi-photon conversion events in the foil.

## Methods

**Experimental setup.** The incoming positron encounters the scintillators S1, S2, and S3, which are used to make the trigger signal, see Figure 1. The positron rate is sufficiently low such that in each event only a single positron enters the setup. The positron then enters a He chamber where the two first position-sensitive (2 cm × 1 cm) MIMOSA-26 detectors are placed. Shortly after the He chamber the crystal target is placed. The He chamber reduces multiple scattering of the positrons, as opposed to using air, such that the incoming particle angle can be measured precisely using the detectors M1 and M2. After the positron enters the crystal, multiple photons and charged particles will leave the crystal. We have ensured also numerically that electron-positron pair production by the emitted photons is negligible in the considered experimental conditions. Indeed, as it is shown in ref. [26] in the table titled: Characteristics of pair production process (p. 270), under the conditions of our experiment, strong-field pair production becomes sizable for photons with energies higher than 150 GeV, which are a negligible fraction in our experiment. To sweep away the charged particles, two large magnets were placed before the final set of tracking detectors. The photons emitted inside the crystal then reach a thin converter foil, 200 μm of Ta, corresponding to ~5% of the radiation length $X_0$, which in turn corresponds to 7/9 of the mean free path for pair production by a high-energy photon[34]. The thickness was optimized such that most of the time a single photon among those emitted by each positron converts to an electron-positron pair. The produced pair then passes through M3 and M4 before entering a small magnet, such that the momenta of the electron and the positron can be determined based on the resulting angular deflection. Finally, the

deflected electron and positron pass through M5 and M6. As we have mentioned, unlike using a calorimeter, this setup has the great advantage that it allows one to measure the single-photon radiation spectrum since only a single, randomly chosen, of the several emitted photons converts to a pair in the thin foil. It is important to point out that for photon energies much larger than the electron rest energy, as most of those emitted in our experiment, the conversion of a photon into an electron-positron pair in the thin foil is independent of the photon energy[34]. Thus, the presence of the thin foil does not alter the spectrum of the photons emitted in the crystal. The tracking algorithm used in the analysis of the data to correctly determine the energy of the photon, which originated from the measured electron and positron is described below. It is clear that the spectrum originating from this procedure cannot be directly compared to the theory since the response of the setup is complicated by experimental effects such as multiple scattering in the converter foil and the presence of air. Therefore, a simulation of the experimental setup, which can translate the theoretical photon spectra into the corresponding experimental ones has been developed, the details of which can be found in Supplementary Note 2.

**Tracking algorithm.** A tracking algorithm has been employed in the analysis of the experimental data in order to correctly characterize the created electrons and positrons, and determine whether they arise from a converting photon in the foil. This is decided based on a series of conditions: hypothetical rectilinear tracks in the detectors M3-M4 and M5-M6 (see Fig. 1) are constructed by connecting all possible pairs of hits in the two planes of M3-M4 and in the two planes of M5-M6. These track candidates in M5-M6 must be matched with those in M3-M4 giving a full particle track, identified by the following conditions.

First, the tracks for individual particles arising from two points in M3-M4 and from two points in M5-M6 are ideally continued into the magnet and, in order to be accepted, they must have a distance to each other within 0.8 mm in the center of the magnet.

Second, the two tracks from the detectors M3-M4 and M5-M6 should be at the shortest distance from each other approximately at the z position of the center of the magnet.

Finally, the size of the deflection angle between the tracks in M5-M6 and the tracks in M3-M4 in the y direction must be smaller than 2 mrad because the magnet deflects only along the x direction, which is the direction orthogonal to z and in the plane of the top view seen in Fig 1.

Now, tracks of electrons and positrons have been individually identified. Moreover, these must also be paired to stem from the same photon. This identification is carried out by requiring that an electron and a positron track must originate from within a distance of 20 μm on the x–y plane in the converter foil. After the identification of the tracks, it may happen that for a given electron or positron, more than one particle of opposite charge matches within the mentioned distance in the converter foil. If this happens, the event is discarded because more than one photon must have converted in the foil and it is not possible to unequivocally associate the electron-positron pair with a photon.

This also implies that if the number of photons above the pair production threshold in the converter foil exceeds ~26, one will begin to see the experimental photon spectrum drop due to multiple photon conversion. We recall that, as we have mentioned in the main text, the thickness of the converter foil corresponds to about $(7/9) \times 5\% \simeq 1/26$ of the average length that a photon covers before converting into an electron-positron pair. Therefore, optimally, this regime is avoided.

In each event all tracks are determined in M1, M2, and M3 as well. The chosen track in these detectors is the one with the closest approach to the pair origin already determined in the converter foil. Finally, the positron entry angle is determined from the hits in M1 and M2 of this track.

**Data availability.** The authors declare that all data supporting the findings of this study are available within the article and its Supplementary Information files or from the corresponding author upon reasonable request.

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

## Acknowledgements

We acknowledge the technical help and expertise from Per Bluhme Christensen, Erik Loft Larsen, and Frank Daugaard (AU) in setting up the experiment and data acquisition. This work was partially supported by a research grant (VKR023371) from VILLUM FONDEN. The authors would like to acknowledge an insightful input from A. Macchi on the different models of radiation reaction.

## Author contributions

T.N.W. and U.I.U. conceived and carried out the experiment with participation of A.D.P. and H.V.K. T.N.W. and A.D.P. carried out the theoretical calculations. T.N.W. carried out the data analysis. T.N.W. and A.D.P. wrote the paper with input and discussion from H.V.K. and U.I.U.
