## [Peer Review File · Nature Communications]

Reviewers' comments:

Reviewer #3 (Remarks to the Author):

I appreciate that the authors have put in considerable efforts to improve their manuscript as documented by their 26 pages reply and the ensuing changes in the manuscript. I'm happy with most of their answers, in particular those regarding synchrotron radiation, which indeed happens at small values of the chi parameter, $\chi \ll 1$, thus in a predominantly classical regime. The reported experiment, on the other hand, has chi of order unity, hence quantum effects will be important and not just a small correction. (This could actually be emphasised a little bit more in the manuscript.)

There are still a few issues, though, which need to be addressed before I can recommend publication in Nature Physics.

1. The definition of quantum radiation reaction remains somewhat muddled and long-winded. This is true for the reply to my previous Remark 1 and particularly so for the abstract, which should generally be streamlined, sharpened and shortened. Why not replace the first sentence by something like "Quantum radiation reaction is the influence of multi-photon emission from a charged particle on the dynamics of the particle itself, characterised by a significant energy-momentum loss per emission"? I believe this is what the authors mean in a nutshell, cf. p.6, line 119, and the answer to Remark 14.

I also find it confusing that further down in the abstract the authors talk about "both quantum and radiation-reaction effects", which suggests these are separate issues in contrast to the "unified" notion of "quantum radiation reaction".

BTW: In line 9, p.2 (5th line of abstract), please replace "physical inconsistent" by "physically inconsistent". In line 29, p.2, I'd say "*along* the same line" rather than "in..." . "Radiation reaction" should generally be written consistently without hyphen.

2. Thanks in part to the statistical analysis, it is now fairly obvious that there is no adequate theory describing the experiment *quantitatively* in a satisfactory manner - which admittedly is a hard problem. As the authors state themselves this is mostly due to a failure of the CCF approximation. The authors should thus be more careful when they say (l.21, p.2) that "only a full quantum theory of radiation reaction is capable of *explaining* (my emphasis) the experimental results". The problem could actually be turned into a virtue: the experiment and its analysis shows that current theory is inadequate and thus more theoretical work is required. I guess this means one should put more emphasis on the experimental findings than on the theoretical analysis.

3. On p.4 (newly added material) all of a sudden the notion of a background field is introduced in the context of QED. However, QED per se has no "background field" - there is no such thing in the QED Lagrangian. I think a few more words of explanation are required here as to why it makes sense to identify the "crystal field" of the abstract with a "background field". Note that this is related to the notion of QRR, which says that the radiation field (= emitted photons) is *not* a small correction to the background field.

4. I repeat that the picture size should be increased, at least upon publication.

Once the appropriate changes have been made, the paper should be suitable for publication in Nature Physics.

Reviewer #4 (Remarks to the Author):

Review by Prof. IC Edmond Turcu sent on 2017.11.10:

Recommendation: Publish manuscript 'as is' on 2017.11.02.

The authors claim to have made the first fundamental experimental test of quantum-electrodynamics [QED] in the regime where the dynamics of charged particles is strongly influenced not only by the external electromagnetic fields but also by the radiation-field generated by the charges themselves and where each photon emission potentially reduces the energy of the charge by a significant amount. The authors claim to have made the first experimental measurement of radiation-reaction emission spectra from ultra-relativistic particles in a strong electric field. The particles are ultra-relativistic positrons crossing the strong electric field of the silicon crystal lattice. It is claimed that only a full quantum theory of radiation reaction is capable of explaining the experimental results, with radiation-reaction effects arising from the recoils undergone by the positrons during multiple photon emissions. It is also claimed that this work opens a new direction to experimental investigations in QED which could also help solve the fundamental question about the structure of the electromagnetic field close to elementary charges.

These claims are truly revolutionary. The experimental observation of QED effects like Radiation-Reaction [R-R] proved elusive so far. This is despite a huge theoretical and experimental effort underway over the last few years. There are whole International Conferences dedicated to the subject (e.g. ExHILP2017 just held in Lisbon, Portugal). Very large, Billion Dollar, experimental installations are being constructed around in Europe and China to observe the QED effects experimentally: the 10PW Laser Projects of ELI in Europe and the 100PW Laser Project of SEL in China. This is a new field of fundamental importance for the understanding of Matter-Light-Field interactions. I think the readers of Nature Journals and the wider community will read these new findings with great interest. The authors write a clear and convincing explanation of the experimental measurements. The experimental arrangement contains only few, well-chosen elements and this leads to a convincing analysis of the measurements. Ultra-relativistic, 178.2 GeV positrons from the CERN accelerator, propagate through Si crystals aligned along the 111 axis. The positron rate is sufficiently low such that in each event only a single positron enters the experimental setup. Gamma-ray photons are emitted from the QED interaction of the ultra-relativistic positrons with the strong electric field inside the Silicon crystal lattice. The Si-crystal is thick as compared to earlier experiments. In order to only measure the Gamma-rays, the positrons are deflected by a magnetic field. In order to measure the Gamma-ray spectrum, the Gamma-rays are converted into electron-positrons pairs in a thin metal foil. The thin foil insures that only a single-photon of the many-photons-emitted/positron is converted into electron-positron pairs. Finally, the momenta of these electrons and positrons are measured by a combination of Magnet and Position-sensitive-detectors. The authors explain that, unlike using a calorimeter, this setup has the great advantage that it allows one to measure the single-photon radiation spectrum. This is a great advantage in the R-R regime where several photons are emitted by a single positron. This experimental arrangement provides more information on the dynamics of the positrons and a stronger test of the theory than previous experimental campaigns.

The data analysis was performed carefully. (1) the emitted Gamma-ray photon spectra were simulated from the measured signal and taking into account the experimental characteristics. (2) Background radiation was measured as shown in Fig.2 –left panel. (3) The validity of the experiment simulation was convincingly tested for Amorphous-Si samples of 3.8mm and 10.0mm length. Fig. 2 –right panel, shows good agreement between measurement and simulation. (4) The experimental background-subtracted Photon Spectra are shown in Fig.3. In Fig.3 the experimental spectra are compared the predictions of four current theoretical models: (i) classical plus radiation reaction model (CRRM), (ii) semi-classical plus radiation reaction model (SCRRM), (iii) quantum plus radiation reaction model

(QRRM), (iv) quantum with no radiation reaction model (QnoRRM). Fig.3.- right-panel shows good agreement between quantum plus radiation reaction model (QRRM) and photon-spectrum measurements for the thick, 10mm, aligned, Silicon-crystal. Fig.3.- left-panel shows that for the 3.8 mm Si-crystal the best agreement is obtained for the classical plus radiation reaction model (CRRM) followed by the quantum plus radiation reaction model (QRRM). In the overall set of data combining the 3.8 mm case and the 10.0 mm case, the QRRM model best describes the experimental data. The conclusion is that the experimental photon-spectra can only be described by a theoretical model which includes both quantum and radiation-reaction effects. The second conclusion is that the theory needs improvement to fully predict the experimental measurement. The chi-squared statistical test (Supplementary Material) show that the disagreement between theory and experiment is not due to statistical-fluctuations. The authors point out that the reason for discrepancy with the quantum plus radiation reaction model (QRRM) is related to the validity of the constant crossed field (CCF) approximation. This is an important conclusion. The experiment tells theorists where to look at improving their theory. In turn the improved theory will improve our understanding of how QED works.

The paper will strongly influence the thinking in the field of QED and Radiation-Reaction. Theorists badly need an experimental demonstration of the Quantum Radiation-Reaction effects. This paper appears to provide such an experiment.

The statistical analysis of the data is appropriate and valid.

The paper provides sufficient experimental detail for a researcher to reproduce the work. Indeed, the CERN positron beamline is available as a user facility. The experimental configuration has few, well characterised elements, easily reproduced.

Dear Reviewers,

Below you find the reports of Reviewer #3 and #4. Since reviewer #4 recommends to publish as is, we answer the criticisms of the Reviewer #3 individually, and reported the corresponding changes in the manuscript and the Supplementary Materials. For the sake of convenience, the additions/changes in the manuscript and in the Supplementary Materials have been highlighted in red.

In summary, we are confident that all remarks have been resolved.

Yours sincerely,

Tobias Nyholm Wistisen, Antonino Di Piazza, Helge V. Knudsen and Ulrik I. Uggerhøj

Response to Reviewer #3

Remark 1

“The definition of quantum radiation reaction remains somewhat muddled and long-winded. This is true for the reply to my previous Remark 1 and particularly so for the abstract, which should generally be streamlined, sharpened and shortened. Why not replace the first sentence by something like Quantum radiation reaction is the influence of multi-photon emission from a charged particle on the dynamics of the particle itself, characterised by a significant energy-momentum loss per emission? I believe this is what the authors mean in a nutshell, cf. p.6, line 119, and the answer to Remark 14.

I also find it confusing that further down in the abstract the authors talk about both quantum and radiation-reaction effects, which suggests these are separate issues in contrast to the unified notion of quantum radiation reaction.

*BTW: In line 9, p.2 (5th line of abstract), please replace physical inconsistent by physically inconsistent. In line 29, p.2, I say *along* the same line rather than in. Radiation reaction should generally be written consistently without hyphen.”*

Answer and corresponding changes in the manuscript:

We agree that the abstract could be refined and have followed the suggestion about the first sentence. We have also changed the sentence regarding “both quantum and radiation-reaction effects” to simply “quantum radiation reaction effects”. We have followed the suggestions on the language formulations and changed “physical” to “physically” (the sentence with “in” to be changed into “along” has been removed, please see below). We have also removed the hyphen from “radiation-reaction” in the whole manuscript and Supplementary Materials.

Moreover, in order to shorten the Abstract, we have removed the last sentence “Future experiments carried out along the same line will be able to, in principle, also shed light on the fundamental question about the structure of the electromagnetic field close to elementary charges.” In fact, in replying to the Remark 2, we have now pointed out the importance of our results for future investigations and we found the above sentence somewhat speculative at the present stage.

Remark 2

*”Thanks in part to the statistical analysis, it is now fairly obvious that there is no adequate theory describing the experiment *quantitatively* in a satisfactory manner - which admittedly is a hard problem. As the authors state themselves this is mostly due to a failure of the CCF approximation. The authors should thus be more careful when they say (1.21, p.2) that only a full quantum theory of radiation reaction is capable of *explaining* (my emphasis) the experimental results. The problem could*

actually be turned into a virtue: the experiment and its analysis shows that current theory is inadequate and thus more theoretical work is required. I guess this means one should put more emphasis on the experimental findings than on the theoretical analysis.”

Answer:

We agree and have followed the suggestion of the referee to replace this sentence of the manuscript (see changes in manuscript below)

Corresponding changes in the manuscript:

We have replaced

”Our theoretical analysis shows that only a full quantum theory of radiation reaction is capable of explaining the experimental results, with radiation reaction effects arising from the recoils undergone by the positrons during multiple photon emissions.”

with

”Our analysis shows that while the widely used quantum approach is overall the best model, it does not completely describe all the data in this regime. Thus, these experimental findings may serve as an inspiration to seek more generally valid methods to describe quantum radiation reaction.”

Remark 3

*”On p.4 (newly added material) all of a sudden the notion of a background field is introduced in the context of QED. However, QED per se has no background field - there is no such thing in the QED Lagrangian. I think a few more words of explanation are required here as to why it makes sense to identify the crystal field of the abstract with a background field. Note that this is related to the notion of QRR, which says that the radiation field (= emitted photons) is *not* a small correction to the background field.”*

Answer:

The notion of “background field” refers to that part of the total electromagnetic field, which is so strong that it does not need to be quantized because it contains a large number of coherent photons and is approximately unaltered during the quantum process. The background field is then treated as a classical field, whose time evolution is given. The effects of the background field are taken into account by quantizing the Dirac field in the presence of that background field, i.e., by solving the Dirac equation in the presence of that background field. In the ultrarelativistic case and when the parameter ξ is much larger than unity it can be shown (see, e.g., [1]) that the effects of an arbitrary background field can be approximately described by means of the constant crossed field approximation.

Corresponding changes in the manuscript:

We have added an explaining sentence such that this part now reads

”Within QED the notion of background electromagnetic field refers to the part of the total electromagnetic field, which is so intense that it does not need to be quantized because it contains a sufficiently large number of coherent photons and is approximately unaltered during the quantum process under investigation [2]. The background field is then treated as a classical field, whose time evolution is given. In the present case the background field is the total field produced by all the atoms of the crystal.”

Remark 4

”I repeat that the picture size should be increased, at least upon publication.”

Answer:

We also agree with this remark of the Reviewer and we have rearranged the last three figures putting the two panels one below the other. In this way, each panel is significantly larger than before.

Corresponding changes in the manuscript:

We have changed the figure size and changed the the text in the manuscript to correctly refer to panels a) or b) instead of left or right panel.

References

- [1] V.N. Baier, V.M. Katkov, and V.M. Strakhovenko. *Electromagnetic Processes at High Energies in Oriented Single Crystals*. World Scientific, Singapore, 1998.
- [2] V.B. Berestetskii, E. M. Lifshitz, and L. P. Pitaevskii. *Quantum Electrodynamics*. Elsevier Butterworth-Heinemann, Oxford, 1982.

REVIEWERS' COMMENTS:

Reviewer #3 (Remarks to the Author):

The authors have addressed all my remaining issues in a satisfactory manner. I'm thus happy to finally recommend publication of the manuscript in Nature Communications.

Dear Reviewers,

Below you find the report of Reviewer #3. Since reviewer #3 recommends to publish as is, we have made no changes in the manuscript based on this, but only made changes to conform to the editorial requirements.

Yours sincerely,

Tobias Nyholm Wistisen, Antonino Di Piazza, Helge V. Knudsen and Ulrik I. Uggerhøj

Complete report of Reviewer #3

The authors have addressed all my remaining issues in a satisfactory manner. I'm thus happy to finally recommend publication of the manuscript in Nature Communications.

Response to Reviewer #3

We have made no changes since Reviewer #3 is now satisfied.